# Laparoscopic Microwave Ablation: Which Technologies Improve the Results

**DOI:** 10.3390/cancers15061814

**Published:** 2023-03-17

**Authors:** Roberto Santambrogio, Maurizio Vertemati, Matteo Barabino, Marco Antonio Zappa

**Affiliations:** 1UOC di Chirurgia Generale, Ospedale Fatebenefratelli, ASST Fatebenefratelli Sacco, 20121 Milano, Italy; 2Department of Biomedical and Clinical Sciences “L. Sacco”, Università degli Studi di Milano, 20157 Milano, Italy; 3Hepatobiliary Surgery Unit, Department of Surgery, San Paolo Hospital, University of Milan, 20142 Milano, Italy

**Keywords:** hepatic surgery, thermoablation, laparoscopy, liver tumor, 3D models, ICG fluorescence imaging, laparoscopic ultrasound

## Abstract

**Simple Summary:**

Laparoscopic ablation of hepatic tumors is a demanding procedure. In this article, Authors show the new technologies which permitted to perform this procedure safely and obtaining good oncological results. In the preoperative period, 3D reconstruction of radiological imaging permits to evaluate exactly the position of the lesions. Intraoperatively, it is possible to guide the treatment using ICG-fluorescence imaging and the intraoperative ultrasound. All these technologies are very useful tools to permit the surgeon to obtain the best results after laparoscopic ablative treatments.

**Abstract:**

Liver resection is the best treatment for hepatocellular carcinoma (HCC) when resectable. Unfortunately, many patients with HCC cannot undergo liver resection. Percutaneous thermoablation represents a valid alternative for inoperable neoplasms and for small HCCs, but it is not always possible to accomplish it. In cases where the percutaneous approach is not feasible (not a visible lesion or in hazardous locations), laparoscopic thermoablation may be indicated. HCC diagnosis is commonly obtained from imaging modalities, such as CT and MRI, However, the interpretation of radiological images, which have a two-dimensional appearance, during the surgical procedure and in particular during laparoscopy, can be very difficult in many cases for the surgeon who has to treat the tumor in a three-dimensional environment. In recent years, more technologies have helped surgeons to improve the results after ablative treatments. The three-dimensional reconstruction of the radiological images has allowed the surgeon to assess the exact position of the tumor both before the surgery (virtual reality) and during the surgery with immersive techniques (augmented reality). Furthermore, indocyanine green (ICG) fluorescence imaging seems to be a valid tool to enhance the precision of laparoscopic thermoablation. Finally, the association with laparoscopic ultrasound with contrast media could improve the localization and characteristics of tumor lesions. This article describes the use of hepatic three-dimensional modeling, ICG fluorescence imaging and laparoscopic ultrasound examination, convenient for improving the preoperative surgical preparation for personalized laparoscopic approach.

## 1. Introduction

Hepatic resections and thermoablation procedures represent important options in the therapeutic strategies for hepatocellular carcinoma (HCC). Very often, they are performed by a laparoscopic approach [1,2] that can be a complicated procedure for the surgeon.

In recent years, several technologies have been developed to facilitate the surgical maneuvers and to improve the results in terms of radicality and complications.

These technologies have evolved simultaneously with the rapid development of laparoscopic surgery in which preoperative and intraoperative planning was more necessary due to accuracy than open surgery [3,4].

This review shows the importance and the limits in the use of some technologies employed in the planning and therapeutic strategies of laparoscopic thermoablation: the hepatic 3D modeling and printing procedure, the indocyanine green (ICG) fluorescence imaging and the laparoscopic ultrasound (LUS) with contrast media. A video describes some applications of these technologies in selected cases (see Appendix A).

## 2. Use of 3D Reconstruction

The complexity of liver vascular anatomy and the anatomic variations and the relationship with focal liver lesions emphasize the importance of surgical planning in liver cancer surgery. In this context, 3D reconstruction has on the one hand increased our level of knowledge regarding the anatomical variants (i.e., 3D reconstruction is tailored on patient-specific anatomy), on the other hand, it allows the possibility of developing new laparoscopic procedures and approaches [5]. Jiang et al. carried out a systematic meta-analysis to compare the difference between 3D reconstruction and 2D CT scans before surgery for primary hepatic carcinoma. They found that preoperative 3D reconstruction has a positive impact on liver surgery reducing damage to liver blood vessels, avoiding intraoperative bleeding during the operation, and achieving the accuracy of the tumor resection [6]. Moreover, 3D reconstruction in conjunction with 2D imaging could be a useful model for improving trainees’ understanding of liver anatomy and surgical resection [7].

During the laparoscopy, to help with surgical maneuvers, the ultrasound view of the spatial relationship of intra-hepatic structures is settled on two-dimensional (2D) images using a high-frequency transducer with a limited depth (about 6 cm) that does not provide a panoramic visualization of the intra-hepatic anatomy [8]. The identification of the exact location of a nodule within the hepatic parenchyma is established by the relationships with the vascular-biliary structures that define the segmental anatomy of the liver and which is very often different in many patients [9,10]. Based on the two-dimensional information obtained from preoperative imaging (CT and MRI), expert surgeons can have a mental picture of a 3D representation scan to perform the operation successfully, but it can be a serious challenge for surgeons to identify the presence of anatomical variants using only the LUS evaluation [11,12]. In this setting, 3D reconstruction from 2D images and virtual reality technologies can clearly show the exact spatial anatomy of a nodule and can help LUS in planning the thermoablation procedures [13,14,15]. Unlike traditional imaging, virtual reality (as well as augmented reality) allows a three-dimensional view of the patient in the form of a copy of the original. This 3D representation increases the visibility of the organs to be examined making them more perceptible in their real form, thus allowing the surgeon to immerse themselves in the image that was created, interact with it, and navigate within its space.

This technology is based on three fundamental principles: immersion, navigation and interaction. The immersion corresponds to the feeling of being “submerged” inside the 3D image. Navigation corresponds to movement within the image. The interaction consists of the manipulation of the structures of the image created in real-time (Figure 1).

The 3D reconstruction procedure is not yet an easy technique and requires different steps before reaching the final product. Of course, a thin-layer, three-phase CT is required. MRI is definitely a more precise imaging technique in the diagnostic definition of the tumor, but 3D reconstruction is more efficient using CT images. In addition, CT should be performed in the three different stages of hepatic perfusion: arterial, portal and late. These phases, indispensable for the diagnostic definition of the nodule, represent the basis for the correct reconstruction of the vascular architecture of the organ and its relationship with the nodule. The result will be improved for the finer (1 mm) layers obtained from the CT images resulting in a more precise and defined 3D reconstruction.

The procedure we use in our Centre is as follows. CT images are loaded and examined with a free, open-source medical image viewer (Horos software Version 3.3.6 or Osirix software version 4.1; Pixmeo, Geneva, Switzerland), designed for Digital Imaging and Communications in Medicine (DICOM) images [16,17]. This software allows the surgeon to perform multiplanar reconstructions of the liver in a very simple way (Figure 2). Such reconstructions are useful for visualizing the lesion, and by analyzing its relationships with contiguous vascular structures, it is possible to identify the exact location of the nodule. In this way, the surgeon is able to program in advance how to place the patient on the operating table in order to have the easiest access to the nodule to be treated.

At this point, the procedure becomes more complex and requires the help of experienced staff. Starting from the DICOM CT datasets, a 3D reconstruction of liver structures (parenchyma, arterial and venous vessels) is obtained using a segmentation procedure. Different methodologies may be used [18,19], but for 3D liver reconstructions, manual segmentation is definitely the preferable one, although it is tedious and requires much more time. In the first step, 3D Slicer is used, a free open-source software for the advanced analysis and processing of medical imaging [20]. With this program, it is possible to obtain a 3D reconstruction of the liver parenchyma and its vascular structures contained within it as well as tumor nodules using semiautomatic algorithms based on in-built region-growing and threshold algorithms in Hounsfield units, with manual adjustment of the boundary to refine little branches of vessels. This human–machine interface allows for the realistic observation of reconstructed structures, which, depending on the needs, can become transparent, ensuring a better visualization of the vascular relationships with the nodule, highlighting only those of interest for the next surgical procedure. The organ can thus be manipulated in real-time.

The 3D model thus obtained is then re-evaluated by a radiologist expert to confirm the correct reproduction of the CT images. Then, it can be exported in STL file format and then adjusted and converted by using Blender v.2.80 (Blender Foundation, Amsterdam, Netherlands), a 3D computer graphics open-source software. Lastly, the converted file is uploaded in a dedicated Virtual Reality Environment (VRE) developed by our team and based on the game engine Unity (Unity Technologies, San Francisco, CA) [21]. The VRE is visualized in a mobile head-mounted display (i.e., Oculus Quest 2) to provide the surgical team with an immersive visualization of the 3D model. Once immersed in the virtual reality environment, the operator can navigate and interact with the 3D reconstruction in an immersive way, can isolate the structures that interest them making the others transparent, and rotate the organ so as to display the nodule from different angles (Figure 3).

In recent years, technological development has allowed access to the world of augmented reality (i.e., Microsoft HoloLens), which can be used to project three-dimensional holograms on the surrounding physical environment, also allowing interaction with holographic objects and spatial tracking.

This technology can be used in the operating room in association with the images of the laparoscopy column monitor and the surgeon can then interact with the 3D model. All of this can be transmitted on an additional monitor allowing other operators to view the same images, thus also playing an educational role [22]. HoloLens is worn by the surgeon without determining any impediments in its action. They can perform the intraoperative ultrasound by comparing the ultrasound images with the 3D model in front of them. The surgeon can interact with HoloLens by moving or zooming in without running the risk of contaminating the operating field. Surgical maneuvers can be stopped at any time to recheck the information that the 3D model can provide. In case the ultrasound or intraoperative images raises doubts about the vascular distribution, the surgeon can compare in real time such images with those of the 3D model and better address the positioning of the antenna or the direction of the dissection slice (Figure 4).

Although 3D reconstructions in the virtual environment based on VR and AR technologies allow liver surgical approaches to be planned by evaluating intrahepatic liver segmental branches and their spatial relationships with the lesion(s), some limitations are still present. Development of VR and AR, for example, is still expensive and time-consuming [23]. Moreover, the soft and deformable nature of the liver parenchyma and the liver movement during operation because of respiratory cycles may determine a change in relationships between the lesion and intrahepatic vascular and biliary structures and an incorrect match between the 3D reconstruction image and the real intrahepatic structures [24]. In this respect, advancements in hardware and tracking technologies (i.e., sensor-based, marker-based, hybrid tracking technologies) with AR systems can increase the matching and mixing of virtual objects (the 3D liver) with the real environment.

Therefore, the use of 3D reconstructions and the development of models for virtual and augmented reality are key aids to:Choosing the patient’s position on the operating table according to the correct location of the nodule;Once the patient is positioned, the choice of the trocar entry point to be used for the laparoscopic probe can be chosen based on the location of the nodule as visualized by virtual reality [25];The technique of lesion centering using the laparoscopic ultrasound probe is totally free-hand. Comparison of images obtained with intraoperative ultrasound and augmented reality allow the surgeon greater accuracy in locating the antenna insertion point in the liver, so that the nodule can be reached with greater precision [14];Finally, in case the intra-hepatic vascular occlusion (IHVO) technique is to be used, 3D reconstruction together with virtual and augmented reality techniques allow the exact individualization of the vessel feeding the nodule, by obtaining a coagulative ablation of the vessel [26].

As indicated at the beginning of this section, all virtual and augmented reality techniques allow the manipulation of the model: in this way they are of valuable help in a situation such as the laparoscopic approach, which does not allow for the easy manipulation of the liver, in addition because of the use of rigid instrumentation. Certainly, the use of robotic surgery with an immersive vision technique and the use of articulated instruments will be of further help in surgical procedures on the liver. Even more so, laparoscopic ultrasound, using linear, high-frequency transducers and therefore with a limited field, may have difficulty in detecting nodules located in deep sites: the 3D reconstruction can tell the surgeon which areas of the liver to examine most carefully for their location [14,25].

However, as already mentioned at the beginning, 3D reconstruction and the creation of virtual models require time and dedicated personnel who are specialized in handling this software. The process of making a 3D reconstruction from DICOM data and transforming it into a virtual and augmented reality model can take more than 2 h, and in centers with a high volume of patients, it can be a problem to accomplish this.

In most cases that come to the attention of centers of liver surgery, the use of CT images with multiplanar reconstructions and possibly the use of 3D reconstruction only on the computer are sufficient to identify the exact location of the lesion and its main vascular relationships. Only in the presence of complex cases, with important vascular abnormalities that make it difficult to understand their relationship with the nodule, might 3D reconstruction with virtual and augmented reality be necessary, as well as the use of 3D printing that in addition to its real use during surgical procedures has an important educational and training role. In fact, the use of models in 3D printing allows a precise liver segmentation to be obtained, thus improving the spatial understanding of the lesions and their relationship with the vascular structures and favoring the professional growth of all the staff [27]. However, the costs and time needed to build these models currently limit their use.

## 3. ICG Fluorescence Imaging

In recent studies, indocyanine green (ICG) fluorescence seems to be a valid tool to increase the safety of liver resection. Some authors [28,29] showed that LUS integrated with ICG fluorescence imaging permitted the identification of HCC nodules: in fact, the use of an LUS probe may be associated with difficulties in visualizing small subglissonean nodules which could be easily identified by ICG imaging. A single dose of ICG (generally 0.5 mg/kg) used for routine liver function tests (with the LiMON device) should be injected 24–36 h before the surgical procedure and it is sufficient to identify nodules using intraoperative fluorescence imaging. Detection of hepatic lesions by ICG fluorescence is determined by the distinction between the tumoral fluorescent aspect and the nonfluorescent remaining liver parenchyma.

Usually, ICG dye is selectively absorbed by hepatocytes through two specific transmembrane transport systems: organic anion-transporting polypeptides (OATP) and sodium-taurocholateco-transporting polypeptides (NTCP). On the other hand, the ICG excretion into the bile ducts is mediated by the activity of canalicular transporters, called multidrug resistance-associated proteins (MRP), that are expressed in the apical part of the hepatocytes [30].

In this setting, ICG fluorescence imaging can be also used to obtain information on nodule characteristics and to help in targeting the lesion;LUS is a fundamental tool for laparoscopic ablation procedures, but it has some drawbacks and difficulties in interpreting images. ICG fluorescence imaging is a promising method for navigation surgery, which allows the limitations of ultrasound examination to be overcome, above all for the subglissonean nodules. In a minimally invasive setting, ICG fluorescence can substitute the tactile feedback of the hand in the presence of soft parenchyma and in some cases, permits the identification of small superficial nodules not identified by the preoperative imaging modalities, completing the LUS staging. In the presence of macronodular cirrhosis and irregular liver surface, it can overcome the LUS difficulties due to the inadequate contact with the liver parenchyma, in the detection of superficial nodules [31,32,33];ICG imaging is very fast and perfectly integrated into the surgical equipment because fluorescent images of hepatic nodules are visualized by simply fixing on the liver surface with the camera and switching the camera system to the near-infrared function;ICG fluorescence imaging could identify different patterns of fluorescence for HCC nodules according to their grade of differentiation [34]: intense and homogenous fluorescence is indicative of well-differentiated HCC, while moderate or poorly differentiated tumor generates partial or rim-type fluorescence;In some procedures used during the laparoscopic thermoablation (IHVO) [26], it is possible to evaluate the ischemic effect of the occlusion of the vessel feeding the lesion. In this case, the ICG injection is performed immediately after the ablation of the vascular pedicle, and it is possible to visualize the area of the liver surface without IGC fluorescence (Figure 5).

However, this technology presents two major drawbacks. The first is the lack of penetration into the parenchyma (up to 8 mm from the liver surface), so, only superficial tumors can be visualized by this technique. On the second, ICG fluorescence could present a relatively high incidence of false-positive rate: benign nodules, biliary hamartomas and nodular hyperplasia can show a well-defined fluorescence [31,35].

## 4. LUS Evaluation

Intraoperative ultrasound (IOUS) has been the gold standard in preoperative staging of focal liver lesions for some years since it allows the recognition of new nodules misconceived to preoperative imaging methods and represents the unique and indispensable tool for the surgeon’s guidance in intraoperative resections and interstitial treatments. The development of minimally invasive techniques and the use of probes dedicated to laparoscopy have allowed the use of intraoperative ultrasound through a less aggressive route, an extremely important factor, especially in such delicate patients as those suffering from cirrhosis of the liver. Laparoscopic ultrasound (LUS) overcomes the two major limitations of the laparoscopic approach, namely limited inspection of the visible surface of the organ and failure in the tactile inspection of structures, and combines the advantages of the mini-invasive way to the remarkable diagnostic capacity of the contact ultrasound, offering an accuracy comparable to that of the IOUS.

However, the LUS technique is not simple and requires considerable experience: the learning curve has not yet been defined with precision [36]. The diagnostic capacity of the LUS depends not only on the skill of the operator but is certainly influenced by the size and depth of the nodule and the ultrasound pattern of the structure of the hepatic parenchyma. In the case of cirrhosis, the irregular surface associated with the impossibility of compressing increased parenchyma of consistency favors the interposition of air between the liver and the transducer, thus altering the quality of the image [37]. In the presence of multiple nodules, the choice of the position of the inlet trocar for the probe can lead to greater difficulty in finding nodules placed in deep and rear positions [38].

In our Centre, we use ultrasound equipment (Arietta V70, Hitachi, Tokyo, Japan) connected to an LUS probe with a flexible tip, 10 mm in diameter and 50 cm in length. A 5–7.5 MHz linear-array transducer was side-mounted near the tip of the shaft. The length of the transducer surface was 38 mm, which produced an image footprint of approximately 4 cm in length and 6 cm in depth. In recent years a laparoscopic probe has become available that allows the use of a contrast agent (Sonovue, Bracco, Italy) during the LUS examination of the liver: the adjunct of contrast enhancement during the intra-operative ultrasound can ameliorate image visibility, the precise diagnosis of new malignant nodules and ablation efficacy after the ablation of HCC nodules.

Usually, it is sufficient to use two 10/12-mm trocar accesses. Preoperative imaging techniques and their 3D reconstructions are essential in determining the position of the patient on the operating bed depending on the location of the nodule. Usually, the position of the patient on the operating bed is supine with the left arm extended: the surgeon is located either on the right side or between the legs of the patient: this is the usual position for localized lesions in the left lobe and in the fourth segment. The same position but with legs closed and the surgeon positioned to the right of the patient is used for anterior sector nodules. Finally, lesions in segments VI and VII require either an oblique position with the right side elevated up to 45 degrees or a left decubitus position with the right arm elevated and across the chest with the surgeon positioned on the right or on the left side of the operating bed.

The access of the LUS probe into the peritoneal cavity is conditioned by the position of the trocars: first, the umbilical port can be chosen for laparoscopic exploration by the camera, and the second trocar for the LUS access could be decided on the information obtained both the preoperative imaging evaluation and the intraoperative conditions as established by laparoscopy visualization.

LUS examination of the liver can be usually accomplished with a direct contact technique in the presence of the natural humidity of the liver surface, which permits an excellent acoustic touch with the transducer. However, for lesions at the hepatic dome or localized in the posterior segments, contact between the transducer and liver surface may be limited: in this case, the saline solution can be introduced between the liver and diaphragm thus creating an acoustic window that allows you to view these areas without air interference (water-immersion method). If, however, this is not sufficient, it is possible to decrease the pneumoperitoneum to 6–8 mm thus reducing the angle of contact with the hepatic surface, thus improving the ultrasound image. With these devices, it is also possible during laparoscopy as with open surgery to completely examine the liver with the ultrasound probe to identify nodules, their size, characteristics and their exact location according to the Couinaud classification of liver anatomy [39].

After the lesions have been localized, the antenna for a microwave ablation (MWA) (or electrode needle for radiofrequency ablation (RFA)) can be accurately inserted into the nodule. When it is necessary to treat lesions localized in segment 1 or in the posterior segments, a longer laparoscopic antenna could be required (27–30 cm). The antenna proceeds through the abdominal wall and the pneumoperitoneum space prior to reaching the liver surface: this space could increase the difficulties in the movements of the tip of the antenna. In fact, laparoscopic ablation is more technically difficult than percutaneous procedure due to a different three-dimensional spatial plan, decreased liberty in antenna angulation and introduction, and with the free-hand guide of intraoperative ultrasound (IOUS) [40]. Other laparoscopic probes have been equipped with a pore for a guide to insert the antenna: however, the rigid direction line due to the pore prevents the possibility of guiding the antenna into lesions located in different sites of the liver [41].

A LUS-guided interventional procedure [42] can be successfully accomplished if the following ideal conditions are satisfied: (1) a good visualization of the lesion is necessary: the ultrasound probe must be positioned on the liver surface to show the largest diameter of the whole nodule; (2) the antenna must be inserted as close as possible to the transducer of the ultrasound probe trying to have an angle oblique to it so that you can follow the path of the antenna tip as closely as possible with the ultrasound. Once the antenna has been inserted, with delicate rotational movements of the probe on its axis, it must follow the path of it up to the lesion. Being a totally free-hand technique, it is possible that the first attempt will not be adequate: in this case, based on the position of the antenna with respect to the nodule, it is extracted by repositioning it at a different point on the liver surface, taking into account that small distances on the surface of the liver can result in much wider distances within the liver, especially if the nodule is very deep. For lesions located in the posterior segments, it is necessary to introduce the antenna on the liver surface distant from the lesion: in this case, the transducer cannot contemporaneously display the nodule and the antenna tip.

A technical variant is the IHVO. This approach has the goal of producing an ischemic zone centered on the vessel feeding the lesion thus increasing the area of necrosis [26]: in this way, it is desired to reduce the risk of partial necrosis or a local tumor progression that would require additional ablative therapy. The first step of the procedure is to detect with ultrasound the presence of the vessel feeding the lesion or otherwise of a contiguous vessel that could determine a heat ink effect: very often the use of color-doppler allows for the better identification of the vascular architecture. At this point, under ultrasound guidance, the electrode/antenna is positioned as close as possible to the vessel and its ablation is carried out by RFA (the ablation cycle lasts either 2–4 min) or MWA (60–90 s). At the end of ablation, the coagulative ablation of the area previously refurnished by the vessel ablated is evaluated using again the color doppler imaging or using intravenous ultrasound contrast agent. Furthermore, a discolored area on the liver surface could be also visualized. The use of ICG imaging after immediate injection can emphasize the absence of portal flow. Then, the nodule can be treated with the insertion of the electrode or antenna in the usual way. IHVO reduces the local recurrence after ablation with comparable results to surgical resection [2,43].

Another application of IOUS is the ability to characterize locally advanced HCC nodules, even with a diameter of less than 3 cm and this could justify the high recurrence rates after radical treatments, mainly due to new HCC tumors in the remaining liver. Several studies have established that the existence of microvascular infiltration and satellitosis could be a determinant of HCC recurrence after healing treatments [44,45,46] and therefore the relevance of recognizing the HCC nodule with these features before the histological analysis [47,48]. In fact, if it was possible to identify the biological aggressiveness of the HCC nodule, a micro-invasive HCC (MI-HCC) could be treated in another way. In a previous study [49], we have shown that IOUS is able to identify some features which were typical of MI-HCC and these findings effectively matched with similar histopathologic criteria. These IOUS findings (Figure 6) are useful to identify the biological behavior of HCC and to foresee the likelihood of local recurrences and overall survival rates. The ability of IOUS to identify the micro-invasive pattern allows for the recognition of these high-risk patients also during laparoscopic ablation, a treatment without a pathological specimen to analyze. In patients with MI-HCC, IOUS (during the laparoscopic ablation treatment) can definitely demonstrate situations at higher risk of local recurrence, secondary to the microvascular infiltration, satellitosis and/or lesions adjacent to the major vessels as described elsewhere [50]. These IOUS’ patterns could be a reason for overlapping needle insertions to secure a larger necrosis area comprehending the territory with the microvascular infiltration and/or satellitosis. In the future, wider use of laparoscopic contrast-enhanced IOUS could improve the success of the laparoscopic ablation treatment thanks to the intraoperative evaluation of the extension of the necrosis area and the presence or not of residual disease. In fact, apart from the capacity of laparoscopic contrast-enhanced IOUS in characterizing new nodules, it can be useful to evaluate necrosis extension after the ablative treatment during the same procedure. Residual viable neoplastic foci can be recognized by the presence of an enhancement portion on the outskirts of the ablated area, permitting a supplementary treatment in the same session. It is necessary to wait for 10–15 min after the first ablation to assess the residual enhancement portion through the contrast-enhanced IOUS because an immediate appraisal of the ablated area is unsatisfactory for the presence of artifacts due to gas formation or cavitation.

## 5. Conclusions

The 3D reconstruction models for virtual and augmented reality and 3D printing of liver anatomy (with the imaging of lesion and its vascular relationship) are interdisciplinary and original processes, which guarantee better pre-surgical planning and guidance during surgical procedures. ICG fluorescence imaging is a promising technology not only as a navigational tool, but also to identify new superficial lesions and to evaluate the efficacy of IHVO. Finally, laparoscopic contrast-enhanced IOUS is useful to characterize new nodules and detect residual malignant vascular tissue. All these technologies are very useful tools to permit the surgeon to obtain the best results after laparoscopic ablative treatments.

## Figures and Tables

**Figure 1 cancers-15-01814-f001:**
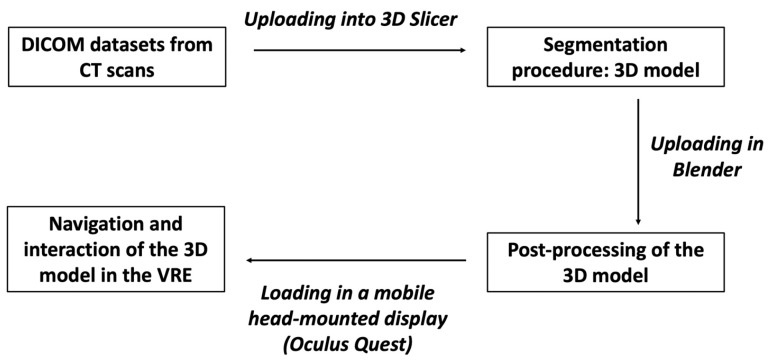
The figure shows the different steps to obtain a 3D model from CT scans to be visualized in the HDMI.

**Figure 2 cancers-15-01814-f002:**
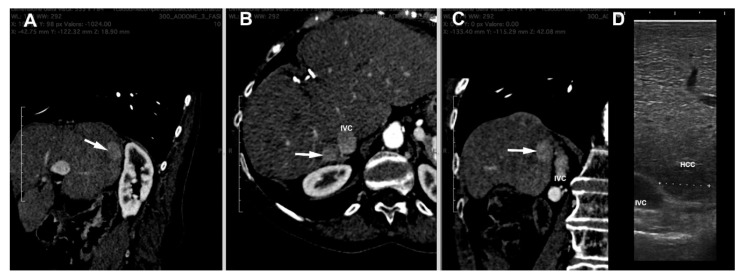
Multiplanar reconstruction of CT scan ((**A**) = sagittal, (**B**) = axial and (**C**) = coronal planes) of a HCC nodule (arrow) in the seventh segment contiguous to inferior vena cava (IVC). (**D**) = laparoscopic ultrasound image of the nodule.

**Figure 3 cancers-15-01814-f003:**
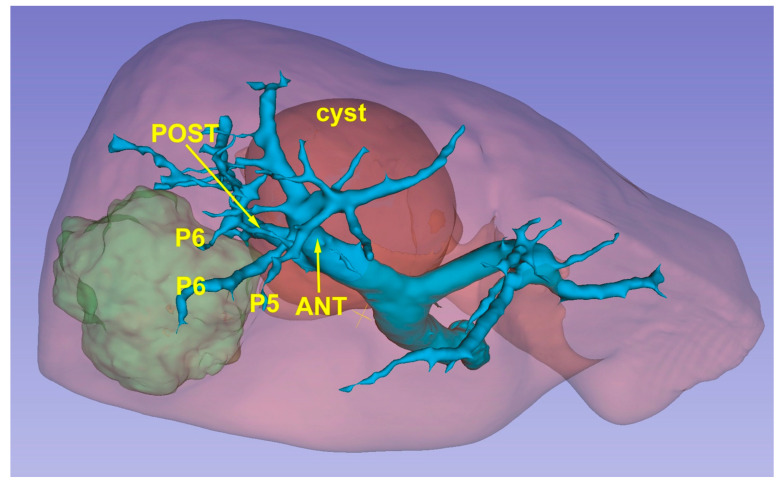
A 3D reconstruction which permits the identification of the lesion’s position (green) in relationship with glissonean pedicles.

**Figure 4 cancers-15-01814-f004:**
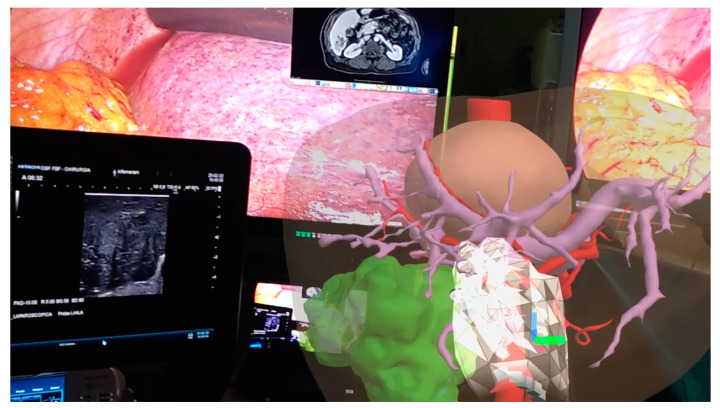
Augmented reality in operating room: surgeon can manipulate the model comparing its features with laparoscopic ultrasound images.

**Figure 5 cancers-15-01814-f005:**
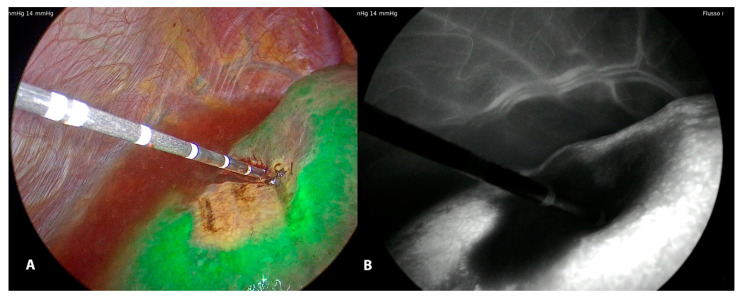
LUS guided puncture to the portal branch feeding the tumor with the fluorescent delineation of vascularized parenchyma (**A**) around the tumor. Monochromatic view (**B**).

**Figure 6 cancers-15-01814-f006:**
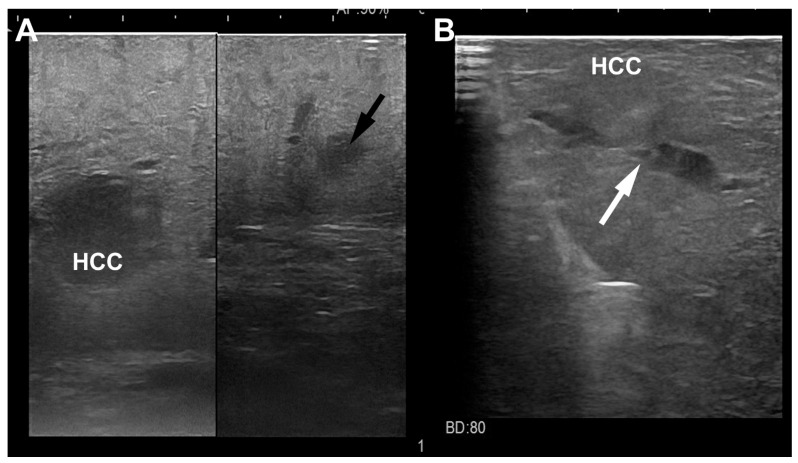
Laparoscopic ultrasound images of a HCC nodule (**A**) located in the sixth segment with a satellite (black arrow) at distance; another HCC nodule (**B**) located in the fifth segment with vascular microinfiltration (white arrow).

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
