# Peer review of "Laparoscopic Microwave Ablation: Which Technologies Improve the Results"

_cancers, 2023, doi:10.3390/cancers15061814_

Round 1

Reviewer 1 Report

This manuscript reviews 3-D imaging, ICG fluorescence imaging, and intraoperative ultrasound in laparoscopic thermoablation for HCC. This review is well written, however, there are some minor drawbacks. I could not access to supplementary materials.

1.       Regarding the application of 3-D imaging to operating room, some figures need to help readers to understand.

2.       Regarding insertion of antenna for ablation, the authors describe the free-hand technique. I think that this technique is often difficult to insert the antenna into tumor accurately. Recently, laparoscopic probe (BK ultrasound), which is equipped with pore for insertion, is sometimes applied to insert the antenna.

Author Response

see file attached

Reviewer 2 Report

Dear Dr. 

Editor, 

Overall recommendation: 

 Accept 

Final comments:

This paper shows the improvement of laparoscopic resection of HCC. Their data looks reasonable. I think this paper is good for good for publication in the present form.  

Kansai Medical University

Katsunori Yoshida

Author Response

REVIEWER 2 ANSWERS

Thank you very much for your evaluation

Reviewer 3 Report

In this manuscript, the authors  review available technologies for safely performing laparoscopic thermal ablation of liver tumors.

In particular, they focus on 3d reconstruction of venous and glissonian intrahepatic intrahepatic structures, on ICG fluorescence imaging, and of LUS.

These are my comments:

-        The manuscript contains many grammar or orthograph inaccuracies (for example, in line 4 of chapter 3, “LUS probe could be some difficulties to visualize….” should be changed to “the use of LUS probe may be associated with difficulties in visualizing….”). In addition, the use of many terms is often not appropriate (for example, line 5 of introduction, “acquired” should be replaced by “developed”). Some sentences lack of punctuation or of a clear organization, making the reading sometimes challenging. Some terms are not correct at all (for example, line 2 of chapter 2, “rapport” should be replaced by ”relationship”). For these reasons the manuscript deserves a deep review by an English mother tongue scientific editor.

-        Chapter 2:

o   Some sentences deliver an unclear or misleading message: for example, at the beginning of chapter 2: the authors state that US does not allow a depth evaluation of liver anatomy, however actually US allows it. In addition, the authors also state that LUS may identify anatomical variations of intrahepatic structures, which may be undiagnosed by preoperative CT scan or MRI; actually, preoperative imaging is very accurate in defining venous  and glissonian intrahepatic anatomy, while LUS may be of help in identifying small tumors that may escape preoperative imaging diagnosis.

o   The description of the procedure for 3d reconstruction is quite long and complex: I suggest the authors to add a figure schematizing the procedure, in order to make the understanding easier.

o   The advantages of the 3d reconstruction are not clear until the end of the chapter, where the authors vaguely state that 3d reconstruction may be of help for complex cases: this aspect should be put at the beginning of the chapter and should be expanded: given the complexity of 3d reconstruction, which are the patients really benefitting from such technique? Why? How the 3d reconstruction may be superior to LUS?

o   A possible limitation of the 3d reconstruction is represented by the soft and deformable nature of the liver parenchyma, which may determine a change in relationships between the tumor and intrahepatic vascular and biliary structures and an inconsistency between 3d reconstruction image and the real intrahepatic structures. This aspect should be discussed in this chapter.

-        Chapter 3:

o   The bulleted list should be reconsidered, given that the different sentences enlisted are not all parts of a list. In addition, such list lack of a description of elements enlisted.

o   Sentence # 2 is basically a repetition of the beginning of the chapter. In addition, it is not clear how ICG, which may give a visual information, may replace the lack of tactile feedback.

Author Response

REVIEWER 3 ANSWERS

  • The manuscript contains many grammar or orthograph inaccuracies
    • I verified the English grammar. In any case, if it is necessary, I can provide to require a check from an editing service
  • Some sentences deliver an unclear or misleading message
    • I changed some sentences regarding the capacity of laparoscopic ultrasound to furnish a panoramic visualization of intra-hepatic anatomy due to the characteristics of the high-frequency transducer. For the comment of preoperative CT or MRI we changed the sentence in: “it can be a serious challenge for surgeons to identify the presence of anatomical variants using only the LUS evaluation [8, 9]. In this setting, 3D reconstruction from 2D images and virtual reality technologies can clearly show the exact spatial anatomy of a nodule and can help LUS for planning the thermoablation procedures”
  • The description of the procedure for 3d reconstruction is quite long and complex
    • According to the suggestion of the reviewer, we have added a Figure (Figure legend: The Figure shows the different steps to obtain from CT scans a 3D model to be visualized in the HDMI)
  • The advantages of the 3d reconstruction are not clear until the end of the chapter
    • A new paragraph has been added at the beginning of chapter 2 explaining the benefits of 3D reconstruction
  • A possible limitation of the 3d reconstruction is represented by the soft and deformable nature of the liver parenchyma
    • a comment has been added to explain this aspect
  • The bulleted list should be reconsidered
    • The bullet list has been removed
  • Sentence # 2 is basically a repetition of the beginning of the chapter

The sentence has been modified to prevent repetitions

Round 2

Reviewer 3 Report

I believe the authors adequately responded to my comments and consistently modified the manuscript, which has been significantly improved. 

However I still believe that a review from an English mother tongue scientific editor is recommended before evaluation for publication. 

Author Response

English has been edited by your editing service
